# High-frequency repetitive transcranial magnetic stimulation at dorsolateral prefrontal cortex for migraine prevention: A protocol for a systematic review of controlled trials

**Nabil Izzaatie Mohamad Safiai**[1], **Nur Afiqah Mohamad**[1], **Hamidon Basri**[1], **Liyana Najwa Inche Mat**[1], **Fan Kee Hoo**[1], **Anna Misyail Abdul Rashid**[1], **Abdul Hanif Khan Yusof Khan**[1], **Wei Chao Loh**[1], **Janudin Baharin**[1], **Aaron Fernandez**[2], **Intan Nureslyna Samsudin**[3], **Mohd Hazmi Mohamed**[4], **Mooi Ching Siew**[5], **Kai Wei Lee**[6], **Vasudevan Ramachandran**[7], **Patricia Pozo-Rosich**[8,9], **Wan Aliaa Wan Sulaiman**[1] *

1 Department of Neurology, Faculty of Medicine and Health Sciences, Universiti Putra Malaysia, Serdang, Selangor, Malaysia, 2 Department of Psychiatry, Faculty of Medicine and Health Sciences, Universiti Putra Malaysia, Serdang, Selangor, Malaysia, 3 Department of Pathology, Faculty of Medicine and Health Sciences, Universiti Putra Malaysia, Serdang, Selangor, Malaysia, 4 Department of ORL-HNS, Faculty of Medicine and Health Sciences, Universiti Putra Malaysia, Serdang, Selangor, Malaysia, 5 Department of Family Medicine, Faculty of Medicine and Health Sciences, Universiti Putra Malaysia, Serdang, Selangor, Malaysia, 6 Department of Pre-Clinical Sciences, Faculty of Medicine and Health Sciences, Universiti Tunku Abdul Rahman, Kajang, Selangor, Malaysia, 7 Bharath Institute of Higher Education and Research, Selaiyur, Chennai, Tamil Nadu, India, 8 Headache and Craniofacial Pain Unit, Neurology Department, Hospital Universitari Vall d'Hebron, Barcelona, Spain, 9 Headache and Neurological Pain Research Group, Vall d'Hebron Research Institute, Barcelona, Spain

* wanaliaa@upm.edu.my

**Funding:** This work is funded by Research Management Centre of Universiti Putra Malaysia under research grant number GPB/2017/9585500.

**Competing interests:** The authors have declared that no competing interests exist.

## Abstract

### Background

Migraine may lead to a negative impact on the patients' quality of life with a subsequent substantial burden to society. Therapy options for treatment and prevention of migraine have progressed over the years and repetitive transcranial magnetic stimulation (rTMS) is one of the promising non-pharmacological options. It induces and alters electric current in the brain via repetitive non-invasive brain stimulation in high frequency. In migraine patients, two common stimulation sites are the M1 cortex and dorsolateral prefrontal cortex (DLPFC). The mechanism on how rTMS exerts therapeutic effects on migraine is not fully established, but the main postulation is that the neuromodulation via high-frequency rTMS (hf-rTMS) might inhibit pain perception. However, evidence from studies has been conflicting, thus the usefulness of hf-rTMS as migraine preventive treatment is still uncertain at this moment.

### Methods

This is a systematic review protocol describing essential reporting items based on the PRISMA for systematic review protocols (PRISMA-P) (Registration number:

CRD42020220636). We aim to review the effectiveness, tolerability, and safety of hf-rTMS at DLPFC in randomised controlled trials (RCTs) as migraine prophylactic treatment. We will search Scopus, Cumulative Index to Nursing and Allied Health Literature Plus, PubMed, Cochrane Central Register of Controlled Trials and Biomed Central for relevant articles from randomised controlled clinical trials that used hf-rTMS applied at DLPFC for the treatment of migraine. The risk of bias will be assessed using the version 2 "Risk of bias" tool from Cochrane Handbook for Systematic Reviews of Interventions Version 6.1. We will investigate the evidence on efficacy, tolerability and safety and we will compare the outcomes between the hf-rTMS intervention and sham groups.

## Discussion

This systematic review will further determine the efficacy, safety, and tolerability of hf-rTMS applied at DLPFC for migraine prophylaxis. It will provide additional data for health practitioners and policymakers about the usefulness of hf-rTMS for migraine preventive treatment.

## Introduction

Migraine is a primary disabling headache disorder [1] that has a negative impact, not only on individuals but also on their families and societies. In 2017, migraine had become one of the most common contributors for disability-adjusted life-years and accounted for about 20% of the burden of neurological disorders in Europe [2].

Fortunately, migraine management has progressed during the last decade. While acute treatment aims to abort or terminate migraine attacks, its use should be limited due to potential development of medication overuse headache. This is where preventive treatment takes an important role as it aims to decrease attack frequency, to improve function and reduce disability [3]. Although pharmacological preventive treatment is the mainstay of practice, adjunctive non-pharmacological treatment, such as neuromodulation, biofeedback, relaxation technique and cognitive behavioural therapy has come into the recent practice with sound recommendations by US guidelines [4].

Since the introduction of transcranial magnetic stimulation (TMS) by Barker in 1985 to stimulate the human motor cortex [5], it has progressed into one of the main non-invasive brain stimulation treatments on patients suffering from neurological and psychiatric diseases [6–9]. The main principle of TMS is passing a large brief current through an insulated coil, generating a magnetic field that transduced through the scalp and generates a weak secondary electrical current that stimulates the brain cells. Later on, repetitive transcranial magnetic stimulation (rTMS) was introduced when advancement in technology allows the generator to produce a rapid pulse in a repetitive manner of high frequency up to 30 Hertz [6].

Currently, the main clinical use of rTMS is in the treatment of depression targeting mainly the dorsolateral prefrontal cortex (DLPFC). The exact mechanism on how rTMS exerts the therapeutic effects on migraine is not yet established. High-frequency rTMS (hf-rTMS) stimulation at DLPFC was found to restore the motor cortical excitability and induce an analgesic effect in capsaicin-induced pain, thus, suggesting that DLPFC plays a role in inhibiting nociceptive transmission [7].

Regarding safety, studies have shown that stimulation was safe with very minimal side effects [8, 9]. A recent review of rTMS applied to pregnant women also found that the stimulation was safe with no reported cases of obstetric complication or severe birth malformations [10]. If the safety is confirmed, this neuromodulation would be an added advantage in managing women with migraine during their pregnancy.

Several systematic reviews had evaluated the use of TMS in headache and migraine [11–14], but none reported the use of hf-rTMS applied at DLPFC in migraine prophylaxis. Therefore, we propose this review protocol to investigate the evidence of the efficacy of the treatment for migraine prophylaxis.

## Methods

### Design and registration

PICO (Population, Intervention, Comparison, Outcome) is used to formulate the research questions [15] of the systematic review. In this review, the population is participants diagnosed with migraine headache. The intervention is hf-rTMS at the DLPFC area, and the comparator is sham stimulation. The main outcome is the treatment efficacy (measured by headache days). The secondary outcomes are tolerability (measured by discontinuation rate) and safety (measured by adverse events and side effects).

In this review, we aim to answer these research questions:

1. What is the efficacy of hf-rTMS at DLPFC in randomised controlled trials (RCTs) in reducing headache days in patients suffering from migraine?

2. What is the tolerability and safety profile of hf-rTMS at DLPFC in randomised controlled trials (RCTs)?

We will follow the recommended reporting items of Preferred Reporting Items for Systematic Reviews and Meta-Analyses (PRISMA) for systematic review protocols (PRISMA-P) (http://www.prismastatement.org). The International Prospective Register of Systematic Reviews registration number for this systematic review is CRD42020220636.

### Data sources and search strategy

Following AMSTAR guidelines, at least two different databases must be searched [16]. To increase more yield, we will search articles from five different data sources: Scopus, Cumulative Index to Nursing and Allied Health Literature Plus, PubMed, Cochrane Central Register of Controlled Trials and Biomed Central.

Based on the Medical subject heading term and previous literature [13, 14], we used the keywords "rTMS" and "repetitive transcranial magnetic stimulation" to identify the rTMS intervention. Meanwhile, the keywords used to identify migraine are migrain*, headache*, hemicran* and migraine disorders [14]. Combining the keywords using the Boolean operator search method, the search strategy is given as an example in Table 1. It will be repeated or modified appropriately for all electronic databases used.

In addition, we will do a few other search methods as described by the Centre for Reviews and Dissemination [17], so that the review will be as inclusive as possible in searching relevant articles. Two trial registries, which are clinicaltrial.gov and the World Health Organization trial registry will be searched too. Besides that, citation searching in which the papers that have cited the included articles will be scanned. We will also look at the reference list of the included studies and other relevant papers to conduct a thorough search for this systematic review.

**Table 1. Search strategy that will be used to retrieve articles from databases.**

| No. | Search terms |
| --- | --- |
| 1. | rTMS or "repetitive transcranial magnetic stimulation" AND migraine* |
| 2. | rTMS or "repetitive transcranial magnetic stimulation" AND headache* |
| 3. | rTMS or "repetitive transcranial magnetic stimulation" AND hemicran* |
| 4. | rTMS or "repetitive transcranial magnetic stimulation" AND migraine disorders |

## Eligibility criteria

**Type of studies.**    This systematic review will include only randomised controlled clinical trials (RCTs) that study migraine treatment using hf-rTMS applied over the DLPFC area in migraine patients. Only articles written in English from inception until December 2020 will be included in this review. During the search process, no language restriction will be applied. However, during the full-text retrieval process, only full-text articles written in English will be included. Conference and proceedings article will be excluded from this review. Studies primarily examining other comorbid conditions with migraine will also be excluded.

**Type of participants, intervention, and comparator.**    Participants of either sex, of any age with migraine diagnosis, will be included in this review. The intervention is specific to hf-rTMS at DLPFC, either left or right DLPFC. Intervention using a different variant of TMS or on other brain areas will not be included in this review. Data will be extracted from RCTs that have both the hf-rTMS intervention and sham stimulation as the comparator.

**Study selection.**    All searches result will be exported to Endnote referencing software, and duplicates will be removed manually. The study screening and selection process will be performed by two independent reviewers. We will do the initial screening using titles and abstracts screening, and those match the interest and relevant to our systematic review will be included. In this process, reviewers will use an inclusive approach rather than an exclusive approach. This process will be documented and summarized in a PRISMA-compliant flow chart (http://www.prisma-statement.org), and the reasons for excluding the studies will be reported.

**Data extraction and quality assessment.**    The process of data extraction will be performed by two independent reviewers. A pre-prepared excel datasheet will be used by the reviewers. Insufficient data will be requested from the trialist whenever possible. For each study, the following information will be extracted: Authors' name, publication year, type of migraine, preventive treatment, group allocation of treatment, number of patients randomised (total and per group), gender and mean age of participant, stimulation protocol, primary outcome and additional outcome, side effects and dropout (and reasons).

For quality assessment, 3–4 independent reviewers will assess the articles using the version 2 Cochrane risk-of-bias tool for randomised trials (RoB 2) from the Cochrane Handbook for Systematic Reviews of Interventions Version 6.1 [18]. The bias that will be assessed includes bias arising from the randomisation process, bias due to deviations from intended interventions, bias due to missing outcome data, the bias in the measurement of the outcome and bias in the selection of the reported result. Disagreements will be resolved by discussion between the reviewers.

**Data synthesis and statistical analyses.**    Data from the intervention will be compared with the data from the comparator sham group. If feasible, a meta-analysis will be performed to determine the most efficacious and tolerable hf-rTMS protocol. Data synthesis will be performed using the latest version of Cochrane Collaboration's software program Review Manager (RevMan) V.5.4.1 for desktop.

For dichotomous data, the outcome will be presented as relative risks (RRs) with 95% CIs. For continuous data, the effect size of the interventions will be calculated using the mean

differences (MDs) with 95% CIs. If the study trials present the outcome values using different scales, the standard mean difference (SMD) with 95% CIs will be used. Meanwhile, the data for the meta-analysis will be calculated using fixed or random effects. If quantitative data synthesis is not possible, a narrative analysis will be performed.

For heterogeneity assessment, the degree of heterogeneity between the studies will be calculated using the $I^2$ statistic. Value >50% will be considered indicative of substantial heterogeneity. If the level of heterogeneity is high, subgroup analysis will be performed to explore the possible causes of heterogeneity [18]. Subgroup analyses will be performed according to factors affecting the outcomes.

## Discussion

Currently, single-pulse TMS on the occipital cortex, has been approved by the Food and Drug Administration for acute pain relief of migraine with aura [19]. On the contrary, the use of rTMS for the preventive treatment of migraine has not been approved yet. A recent INS-NANS 2020 expert consensus panel had graded hf-rTMS at M1 for migraine prevention as having a moderate level of certainty regarding net benefit based on the currently available evidence strength. However, hf-rTMS at DLPFC, low-frequency rTMS at a vertex, continuous theta bursts stimulation at M1 were graded as having a low level of certainty [20].

A systematic review of hf-rTMS on motor cortex areas in migraine preventive treatment has been and its results demonstrated the effectiveness of hf-TMS for migraine prevention [13]. This can be an evidence to support the use of rTMS on the M1 cortex as one of the preventive migraine treatments. To our knowledge, there is still no systematic review of hf-rTMS at DLPFC for migraine to date. Thus, its effectiveness for migraine prevention needs to be investigated [11, 21].

The present systematic review protocol will assess the efficacy, safety, and tolerability of hf-rTMS applied at DLPFC for migraine preventive treatment. The result will provide useful data for health practitioners and health policymakers in extending the clinical evidence of hf-rTMS at DLPFC for migraine preventive treatment.

## Supporting information

**S1 Checklist. PRISMA-P (Preferred Reporting Items for Systematic review and Meta-Analysis Protocols) 2015 checklist: Recommended items to address in a systematic review protocol**[*].
(DOC)

## Author Contributions

**Conceptualization:** Nabil Izzaatie Mohamad Safiai, Liyana Najwa Inche Mat, Mohd Hazmi Mohamed, Wan Aliaa Wan Sulaiman.

**Data curation:** Nabil Izzaatie Mohamad Safiai, Nur Afiqah Mohamad, Kai Wei Lee.

**Formal analysis:** Nabil Izzaatie Mohamad Safiai, Fan Kee Hoo, Anna Misyail Abdul Rashid, Abdul Hanif Khan Yusof Khan, Wei Chao Loh, Janudin Baharin, Mooi Ching Siew, Kai Wei Lee.

**Funding acquisition:** Hamidon Basri, Liyana Najwa Inche Mat, Fan Kee Hoo, Aaron Fernandez, Intan Nureslyna Samsudin, Mohd Hazmi Mohamed, Mooi Ching Siew, Wan Aliaa Wan Sulaiman.

**Methodology:** Nabil Izzaatie Mohamad Safiai, Nur Afiqah Mohamad, Vasudevan Ramachandran, Patricia Pozo-Rosich.

**Project administration:** Hamidon Basri, Aaron Fernandez, Intan Nureslyna Samsudin.

**Supervision:** Hamidon Basri, Aaron Fernandez, Intan Nureslyna Samsudin, Wan Aliaa Wan Sulaiman.

**Writing – original draft:** Nabil Izzaatie Mohamad Safiai.

**Writing – review & editing:** Nabil Izzaatie Mohamad Safiai, Nur Afiqah Mohamad, Vasudevan Ramachandran, Patricia Pozo-Rosich, Wan Aliaa Wan Sulaiman.

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
