## [Decision Letter · Decision Letter 0]

28 Apr 2021

High-frequency repetitive transcranial magnetic stimulation at dorsolateral prefrontal cortex for migraine prevention: A protocol for a systematic review of controlled trials

PONE-D-21-07015

Dear Dr. Wan Sulaiman,

We’re pleased to inform you that your manuscript has been judged scientifically suitable for publication and will be formally accepted for publication once it meets all outstanding technical requirements.

Kind regards,

Sherief Ghozy, M.D., Ph.D. candidate

Academic Editor

PLOS ONE

Reviewers' comments:

Reviewer's Responses to Questions

**Comments to the Author**

1. Does the manuscript provide a valid rationale for the proposed study, with clearly identified and justified research questions?

Reviewer #1: Yes

2. Is the protocol technically sound and planned in a manner that will lead to a meaningful outcome and allow testing the stated hypotheses?

Reviewer #1: Yes

3. Is the methodology feasible and described in sufficient detail to allow the work to be replicable?

Reviewer #1: Yes

4. Have the authors described where all data underlying the findings will be made available when the study is complete?

Reviewer #1: Yes

5. Is the manuscript presented in an intelligible fashion and written in standard English?

Reviewer #1: Yes

6. Review Comments to the Author

You may also provide optional suggestions and comments to authors that they might find helpful in planning their study.

Reviewer #1: The manuscript provide a valid rationale for the proposed study, with clearly identified and justified research question. The protocol is technically sound and planned in a manner that will lead to a meaningful outcome and allow testing the stated hypotheses. Also, the methodology is feasible and described in sufficient detail to allow the work to be replicable. There is a statement from the authors to make all data underlying the findings described in their manuscript fully available without restriction when the study completed. The language is clear and correct.

* About the main outcome which is the treatment efficacy (measured by headache days), why do not you include the headache intensity/severity and frequency of attacks to define the treatment efficacy?

7. PLOS authors have the option to publish the peer review history of their article (what does this mean?). If published, this will include your full peer review and any attached files.

Reviewer #1: No

---

## [Editor Report · Acceptance letter]

10 Jun 2021

PONE-D-21-07015 

High-frequency repetitive transcranial magnetic stimulation at dorsolateral prefrontal cortex for migraine prevention: A protocol for a systematic review of controlled trials 

Dear Dr. Wan Sulaiman:

I'm pleased to inform you that your manuscript has been deemed suitable for publication in PLOS ONE. Congratulations! Your manuscript is now with our production department. 

Kind regards, 

on behalf of

Dr. Sherief Ghozy 

Academic Editor

PLOS ONE